# The Late Asymptomatic and Terminal Immunodeficiency Phases in Experimentally FIV-Infected Cats—A Long-Term Study

**DOI:** 10.3390/v15081775

**Published:** 2023-08-21

**Authors:** Brian G. Murphy, Diego Castillo, Sarah Cook, Christina Eckstrand, Samantha Evans, Ellen Sparger, Chris K. Grant

**Affiliations:** 1Department of Pathology, Microbiology and Immunology, School of Veterinary Medicine, University of California, Davis, CA 95616-5270, USA; ldcastillo@ucdavis.edu; 2Specialty VetPath, 3450 16th Avenue W, Suite #303, Seattle, WA 98119, USA; sestevens@ucdavis.edu; 3Department of Veterinary Microbiology and Pathology, College of Veterinary Medicine, Washington State University, Pullman, WA 99164-7034, USA; chrissy.eckstrand@wsu.edu; 4Department of Microbiology, Immunology, and Pathology, College of Veterinary Medicine and Biomedical Sciences, Colorado State University, Fort Collins, CO 80523, USA; samantha.evans@colostate.edu; 5Department of Medicine and Epidemiology, School of Veterinary Medicine, University of California, Davis, CA 95616-5270, USA; eesparger@ucdavis.edu; 6Custom Monoclonals International, 813 Harbor Boulevard, West Sacramento, CA 95691, USA; ckgrantcmi@vetmabs.com

**Keywords:** cat, FIV, lentivirus, lymphoma, chronic renal failure

## Abstract

Feline immunodeficiency virus (FIV) is a lentivirus in the family Retroviridae that infects domestic cats resulting in an immunodeficiency disease featuring a progressive and profound decline in multiple sets of peripheral lymphocytes. Despite compelling evidence of FIV-associated immunopathology, there are conflicting data concerning the clinical effects of FIV infection on host morbidity and mortality. To explore FIV-associated immunopathogenesis and clinical disease, we experimentally inoculated a cohort of four specific pathogen-free kittens with a biological isolate of FIV clade C and continuously monitored these animals along with two uninfected control animals for more than thirteen years from the time of inoculation to the humane euthanasia endpoint. Here, we report the results obtained during the late asymptomatic and terminal phases of FIV infection in this group of experimentally FIV-infected cats.

## 1. Introduction

Feline immunodeficiency virus (FIV) is a naturally occurring lentivirus that infects domestic cats and is associated with life-long viral persistence and a progressive immunopathology. FIV infection of domestic cats is an important animal model of human immunodeficiency virus-1 (HIV-1) pathogenesis [1,2,3], and these two viruses are phylogenetically related [4].

FIV infection of cats is characterized by three sequential stages, including an early acute viremic stage, a prolonged asymptomatic phase, and a terminal immunodeficiency stage [1]. The acute phase begins 1–4 weeks after initial FIV infection, may span a time period of 2–6 months [5], and is characterized by fever, diarrhea, and generalized lymphadenopathy [6]. Although readily detectable during the initial acute phase of infection, plasma viremia becomes generally undetectable during the asymptomatic phase of infection [7], which can last for multiple years or for a significant proportion of the remaining life span of the cat [5,6]. Although there is a progressive decline in peripheral blood CD4 T cells throughout the asymptomatic phase, infected cats generally remain clinically healthy [5,8,9]. The literature suggests that FIV-infected cats in the terminal phase of infection develop a variety of symptoms, including reemergence of generalized lymphadenopathy (lymphadenomegaly), severe wasting, opportunistic infections (cryptococcosis, toxoplasmosis, and a variety of viral infections), neoplasia (especially lymphoma), neurological abnormalities, anemia, and leukopenia [5,10]. FIV infection has also been associated with various diseases of the oral cavity [11,12,13]. Viral loads have been reported to increase during the terminal phase, and survival time is usually less than a year after onset [5].

There is controversy regarding the clinical relevance of FIV in naturally infected cats [14] as some investigators believe that FIV itself does not cause severe clinical disease, and FIV-infected cats live many years without any health problems [15]. Early epidemiological studies suggested environmental factors such as indoor housing versus outdoor free-roaming status impacted FIV infection status [16]. Environmental setting was suggested to be a major factor in disease outcome in a later published report describing two feline cohorts that differed by cat density, i.e., less than two cats per household versus a sixty-cat household [17]. In a recent single institution study of mortality in 3108 cats that underwent a postmortem examination, FIV status was determined to not be associated with decreased longevity [18]. This clinical disconnect between experimental and naturally acquired FIV infection has not been adequately explained. Importantly, although the acute and early asymptomatic stages of experimental FIV infection have been extensively studied, the terminal stage of experimental infection has not, possibly because of the expense of maintaining a group of experimentally infected animals for protracted periods of time.

We experimentally infected a cohort of four specific pathogen-free (SPF) cats with a biological isolate of FIV clade C and serially monitored these animals, along with two uninfected control cats, in an SPF feline research facility at the University of California, Davis, for more than thirteen years from the time of inoculation to death. Three of the infected cats became FIV progressor animals, while one became a long term non-progressor (LTNP) animal, immunologically indistinguishable from the uninfected control cats. We found that FIV establishes a latent infection in peripheral CD4 T cells and an active infection in circulating monocytes [7]. We also found that CD4 T cell latency was associated with epigenetic modification of histone proteins physically associated with the FIV 5′ long terminal repeat (LTR, viral promoter) [19,20]. Despite the absence of detectable viral replication within circulating CD4 T cells, peripheral CD4 T cell numbers precipitously declined in the progressor animals [21,22]. We also identified a progressive loss of both CD8 T cells and CD20 lymphocytes [21] and a concurrent increase in CD11b monocytes [22].

In our cohort of experimentally infected progressor cats, the proviral LTR and *gag* sequences derived from peripheral leukocytes were determined to be unstable over time, despite low to undetectable viral replication in the peripheral blood [23]. No variation from the inoculating viral sequence was identified in the proviral LTR derived from CD4 T cells isolated from the peripheral blood and lymph nodes of the LTNP cat [23]. A survival study focused on surgical biopsies of a central lymph node (LN), spleen, and small intestine revealed that these tissues supported ongoing viral replication, despite highly restricted viral replication in peripheral blood [22].

Serial immunologic and virologic monitoring of each animal continued until predetermined clinical criteria were met, requiring humane euthanasia. At the terminal phase of disease, the cats eventually succumbed to either lymphoma or chronic renal failure (CRF). The single LTNP cat outlived the three progressor animals. The LTNP cat was eventually euthanized for chronic renal disease that was clinically indistinguishable from one of the uninfected control animals. None of the FIV-infected cats showed any evidence of terminal opportunistic infections.

## 2. Materials and Methods

### 2.1. Animals and Pathology

Six FIV specific pathogen-free (SPF) kittens were purchased in 2009 from the breeding colony of the Feline Nutrition and Pet Care Center, University of California, Davis (UC Davis). At the time of purchase, kittens ranged in age from 4 to 5 months and were housed in the Feline Research Laboratory of the Center for Companion Animal Health, UC Davis. Four kittens were intramuscularly inoculated with FIV-C-Pgmr viral inoculums (kittens 165, 184, 186, and 187), and two control kittens (183 and 185) were mock-inoculated intramuscularly with 1 mL of sterile culture media and monitored as previously described [13]. The FIV-C-Pgmr biological isolate was provided by Drs. E. Hoover (Colorado State University) and N. Pedersen (University California, Davis). The experimental study protocols were approved by the UC Davis Institutional Animal Care and Use Committee (IACUC 21706). The cats were monitored and housed as two separate cohorts in an SPF facility (FIV-infected/uninfected, Feline Research Laboratory) throughout their lifespan. The rooms had multiple perches and a variety of cat toys and boxes for enrichment, play, and exercise. The cats were fed and observed, and litter boxes were changed daily. Room lighting cycles were seasonally adjusted.

Periodic physical examinations, chemistry panels, complete blood counts, and urinalyses were performed depending on clinical signs. Humane criteria for euthanasia were established by the principal investigator (Murphy) in collaboration with Campus Veterinary Services and approved by IACUC. The euthanasia criteria included a combination of the following: severe clinical signs for at least one month that were not responsive to treatment, prolonged anorexia resulting in greater than 20% loss of optimal body weight, a hematocrit less than 15% for two consecutive measurements, dehydration requiring fluid therapy 3 or more times in a 7-day period, or a hematological malignancy. A goal was to not prematurely euthanize any cat for a disease process determined to be treatable or recoverable. Euthanasia was performed by intravenous administration of phenytoin sodium/pentobarbitol sodium (Beuthanasia, Merck, Kenilworth, NJ, USA) administered at >100 mg/kg body weight. A complete necropsy and tissue collection was performed for each cat within 1–5 h of euthanasia.

Gross lesions were identified and described during the necropsy examination, and a complete set of tissues was collected and fixed in 10% buffered formalin for microscopic examination. In addition, two additional sets of fresh tissue samples were collected, including brain, lymph node, bone marrow, small intestine, and spleen (<1 g tissue per sample). Tissues from one of these sets were packaged into individual Whirl-Pac bags (Nasco, Toronto, ON, USA) and archived at −80 °C. Tissues from the other sample set were placed into individual sterile microcentrifuge tubes containing 1.5 mL RNAlater (Thermo Fisher Scientific, Waltham, MA, USA) and frozen at −20 °C. Formalin-fixed tissues were trimmed, placed in cassettes, routinely paraffin embedded, and processed for 5 μm-thick sections stained with hematoxylin and eosin. All resulting histopathology slides were examined by veterinary anatomic pathologists (Murphy, Eckstrand, Cook).

### 2.2. Isolation and Enumeration of Peripheral Leukocytes

Monthly peripheral blood mononuclear cells (PBMCs) and plasma were harvested from whole blood collected in EDTA or heparin processed by Ficoll-Hypaque (Sigma, St. Louis, MO, USA) density gradient centrifugation. Plasma was archived at −80 °C until later analysis. The total number of peripheral white blood cells (total WBC) was serially determined as described previously, using either an automated Coulter Counter (Coulter ACT Diff, Beckman Coulter, Brea, CA, USA) or LeukoCheck Kit (Biomedical Polymers, Inc., Gardner, MA, USA) and manual hemocytometer (Hausser Scientific, Horsham, PA, USA) [24,25]. The relative proportions of specific peripheral leukocyte subsets were determined by flow cytometry from 100 μL of whole blood, as described previously [25]. This procedure utilized the following antigen-specific antibodies, anti-feline CD4 (clone FE1.7B12), anti-feline CD8 (clone FE1.10E9), and anti-canine CD21 (B cells, clone CA2.1D6). All of the antibodies were obtained from the laboratory of Dr. P. Moore (UC Davis). Absolute cell counts were calculated by multiplying the total WBC count by the percent of cells expressing the specific antigen marker. CD4 T cells were isolated from feline PBMC by immunomagnetic columns, as described previously [7].

### 2.3. Isolation, Quantification, and Sequencing of Viral Nucleic Acids

Serial attempts were made to isolate and amplify viral RNA from clarified plasma using real-time RT-PCR, as described previously [25]. Quantification of plasma viral RNA was based on a standard curve generated from viral transcripts, prepared by in vitro transcription of a plasmid (pCR2.1, Invitrogen, Carlsbad, CA, USA) containing a 101-nucleotide FIV-C *gag* amplicon [7]. The amount of FIV gag RNA was expressed relative to volume of blood (mL).

DNA and RNA were isolated from frozen abdominal (mesenteric) LN archived in RNAlater at the time of necropsy examination and stored at −20 °C until utilized. For LN, approximately 30 mg of tissue was thawed on ice and mechanically disrupted using a disposable Closed Tissue Grinder System (02-542-09, Fisher scientific, Waltham, MA, USA) in Buffer RLT with β-mercaptoethanol (Qiagen, Hilden, Germany). The disrupted tissue was subsequently homogenized using a QIAshredder column (Qiagen), and RNA was isolated with the RNeasy Kit (Qiagen) according to manufacturer’s instructions. Tissue-associated RNA was DNAse-treated and reverse-transcribed to cDNA, as described previously [7]. The number of cell-associated viral DNA and viral RNA (cDNA) was quantified and normalized to cellular GAPDH via real-time PCR, as described previously [7]. The real-time PCR assay has a detection limit of approximately 10 copies of FIV *gag*-complementary DNA (cDNA) per tissue sample, or 8 × 10^2^ copies of FIV *gag* cDNA per mL plasma [23].

A proviral subgenomic fragment containing the long terminal repeat (LTR) and first ~1000 nucleotides of the FIV leader, *gag capsid* (CA) and 5′ terminus of *gag matrix* (MA), were PCR amplified from genomic DNA isolated from PBMC, peripheral CD4 T cells, and LN and cloned using a commercial system (pCR2.1 TA cloning system, Invitrogen) [25]. Plasmid DNA was purified using a commercial kit (Promega) and sequenced by a local vendor (Davis Sequencing, Davis, CA, USA).

Viral sequences were analyzed for single nucleotide polymorphisms (SNPs), insertions, and deletions relative to the inoculating viral sequence. Viral sequences were determined at multiple time points throughout the infection. Nucleotide sequence of the initial FIV-C-Pgmr virus was determined from cDNA generated from inoculum viral RNA. Viral sequences were aligned and compared using MacVector 18.0 software (MacVector, Inc., Apex, NC, USA).

### 2.4. Antibody Studies

Serial plasma samples were isolated by centrifugation from anticoagulant-treated whole blood (EDTA or heparin) and archived at −80 °C until utilized for serological assays using enzyme-linked immunosorbent assays (ELISA), as described below. Feline plasma was assessed for (i) immunoglobulin specificity to a variety of FIV antigens and the viral host cell receptor; and (ii) FIV-specific immunoglobulin isotypes.

### 2.5. Specificity to FIV Antigens and CD134 Host Cell Receptor

Three FIV antigens (Gag p24, Env SU, and Env TM) and also the primary FIV binding receptor, CD134 (OX40), were individually coated onto the wells of a 96-well ELISA plate (all antigens were obtained from Custom Monoclonals, Sacramento, CA, USA). Antigens were diluted in coating buffer (0.1 M sodium carbonate, pH 9.6) at 3.0 µg/mL (p24) or 1.5 µg/mL (SU, TM and CD134), respectively. In total, 100 µL of each antigen solution was aliquoted into each well, and the plate was incubated at room temperature (RT) overnight. Coated wells were washed twice with PBS-T wash solution (1 L phosphate buffered saline, 1 mL tween 20), and 100 µL of feline plasma was added to each well to a final dilution of 1:300 in PBS-T/BSA (28 mL PBS/T, 2 mL 7.5% bovine serum albumin). Plates were incubated 45 min at RT and washed three times with PBS-T. A total of 100 µL of monoclonal anti-cat IgG (GPB2-2B1) was added at 1.0 µg/well diluted in PBS-T/BSA, incubated 45 min at RT, and washed 3 times with PBS-T. A total of 100 µL of goat anti-mouse IgG-HRP (BioRad) was added at 1:1000 dilution in PBS-T/BSA to each well, incubated 45 min at RT, and washed 3 times with PBS-T. The plate was developed with the horse radish peroxidase (HRP) substrate and o-phenylenediamine dihydrochloride (OPD, Thermo Scientific, Waltham, MA, USA) using standard protocols and 12 min of incubation at RT. The reaction was stopped with 2 M H_2_SO_4_ and scanned for absorbance at 493 nm with a plate reader. One hundred optical density units of background was subtracted from the reading of each well.

### 2.6. FIV-Specific Immunoglobulin Isotypes

To determine the relative amount of isotype-specific anti-FIV antibody in each plasma sample, a set of ELISA reactions was conducted with plates coated with FIV virus, and a small established group of murine anti-cat isotype specific monoclonal antibodies was utilized [26]. Target FIV virus was harvested from Crandell-Rees Feline Kidney (CRFK) cells infected with FIV-petaluma molecular clone. The virus-containing culture supernatant (grown in the transient absence of fetal calf serum) was centrifuged, cell-free supernatant harvested, diluted 1:2 in coating buffer (0.1 M sodium carbonate, pH 9.6), and 100 µL/well applied to an ELISA plate (Santa Cruz Biotechnology, Santa Cruz, CA, USA) and incubated overnight at 4 °C. Virus-coated wells were washed twice with PBS-T and feline plasma samples added to each well to a final dilution of 1:300, as described above. Plates were incubated 45 min at RT, washed three times, and 100 µL of murine anti-isotype antibody was added per well (FDG1-2A1 anti-IgG, CAG8-78C anti-IgG kappa, IgA9-6A anti-IgA, or CM7 anti-IgM, all from Custom Monoclonals, Sacramento, CA, USA) at 1.0 µg/well and incubated for one hour at RT. Wells were then washed, treated with goat anti-mouse HRP conjugate, developed with OPD reagent, and scanned for absorbance with an ELISA plate reader, as described above.

### 2.7. Statistical Tests

Graphical numerical data are presented as the mean of three or more values, with the standard deviation or range represented by error bars. Statistical differences were determined by unpaired Student’s *t*-tests. A *p*-value of <0.05 was considered to be statistically significant. Statistics were performed with Prism 6 software (GraphPad Software, Inc., La Jolla, CA, USA).

## 3. Results

### 3.1. Clinical Timeline

The clinical timeline for all six of the study animals, including four FIV-infected cats (165, 184, 186, and 187) and two uninfected control cats (183 and 185), is depicted in Figure 1. The acute phase of infection (blue box) featured variably evident lymphadenomegaly and lasted for approximately 6–10 months post-infection, while the asymptomatic phase (during which cats were essentially clinically normal) lasted for the majority of the lifespan of the FIV-infected cats (7.5–12.5 years). In contrast, the terminal stage of infection was comparatively brief, lasting for approximately 2–12 weeks, depending on the individual animal. Features of the terminal phase of infection varied by the individual animal but included variable anorexia, weight loss, polyuria, polydipsia, paraparesis (hind leg paralysis), persistent fever, profound leukopenia, azotemia, and progressive nonregenerative anemia. In all cases, IACUC criteria for euthanasia were met prior to euthanasia. The three FIV progressor animals, 165, 186, and 184, were all euthanized prior to the non-progressor animal, cat 187.

The uninfected control cat 183 was euthanized at 11.8 years due to complications of chronic renal failure (CRF), while the remaining control cat 185 was adopted out of the colony after euthanasia of cat 187. Cat 185 was eventually euthanized for a severe and undetermined clinical problem at the age of 15 years.

### 3.2. Experimental FIV Infection Results in a Progressive Loss of Peripheral CD4, CD8, and CD21 Cells

The absolute number of peripheral CD4 T cells for each animal is plotted as a function of time in Figure 2a, and for CD8 T cells in Figure 3a. For the progressor animals 165 and 186, CD4 and CD8 T cells progressively and profoundly decreased throughout the asymptomatic phase of infection. However, for progressor cat 184, although CD4 T cells profoundly decreased from ~200 weeks post-infection (PI) onward, CD8 T cells initially decreased from approximately 340 to 425 weeks PI and then returned to the normal range (500–2500 cells/µL blood) from 450 weeks PI until euthanasia at 12 years of age (Figure 3a). It should be noted that cat 184 lived longer than the other two progressor animals—3.8 years longer than cat 165 and 1.2 years longer than cat 186. Whether the peripheral CD8 T cell rebound had any role in this discrepancy in longevity remains speculative.

The LTNP animal, cat 187, had oscillating CD4 and CD8 lymphocyte numbers that were comparable to the two uninfected control animals, 183 and 185. In aggregate, the mean absolute numbers of peripheral CD4 and CD8 lymphocytes for the progressor animals were significantly less than the two control cats, whereas the average numbers of cells for the LTNP and control cats were similar (Figure 2b and Figure 3b).

Lifetime aggregated data for peripheral CD21 cells and total white blood cells indicated that the FIV progressor cats had significantly fewer cells than the control animals, while the averages were similar between the control and LTNP cats (Figure 4a,b).

### 3.3. Plasma Viremia Is Intermittently Detectable in the Acute and Late Asymptomatic Phases of Infection

All the FIV-infected cats had RNA intermittently detected in the blood plasma (viremia) during the acute phase of infection (first 40 weeks post-infection) that became undetectable from 40–470 weeks PI (early to mid-asymptomatic phase, Figure 5). During the late asymptomatic phase onward (>471 weeks PI), plasma viremia was intermittently detectable in blood samples from each of the remaining FIV-infected cats (184, 186, 187) and ranged from 1 × 10^3^ to 3 × 10^6^ copies *gag* vRNA/mL of blood. For progressor cat 165, plasma viremia was not detected at any time point after 18 weeks PI.

### 3.4. The Proviral LTR and 5′ Gag Sequences Derived from PBMC and Tissues Are Genetically Unstable over Time

The proviral LTR region and first ~1000 nucleotides of the *gag* gene were PCR amplified, cloned, and sequenced from DNA isolated from PBMC and lymph node tissue derived from all the FIV-infected cats. Numerous SNPs (yellow highlight) were identified scattered throughout the U3, R, and U5 regions of the LTR, including within many of the recognized transcription factor binding sites—AP4, AP1, C/EBP, and NF1—and the TATA box (Figure 6). One SNP was also identified within the LTR sequence of the inoculating virus U5 region (blue highlight). Some of the SNPs within the LTR were identified only once, while others were isolated and identified at numerous time points in multiple animals (short black arrows with asterisks). SNPs delineated by black arrows with an asterisk have been previously cloned into expression constructs, and their promoter activity has been assessed [7,23]. A single 5 bp deletion was identified at the 3′ end of U5 region (green highlight).

Within the 5′ portion of proviral *gag* (primer binding site (PBS), bipartite packaging signal, *gag MA*, and *gag CA* genes), multiple SNPs (21, yellow), deletions (8, green), and insertions (3, purple) were identified in the four FIV-infected cats (Figure 7). SNPs and indel mutations were identified within sequences known to be important for efficient encapsidation (underlined region) and scattered within the *matrix* and *capsid* ORFs. No mutations were identified within the splice donor sequences in any cat (grey highlights).

### 3.5. Serologic Responses to FIV Antigen Are Primarily IgG Isotype-Based

Over the course of the infection, all of the FIV-infected cats mounted a relatively strong but variable humoral response to FIV Env (SU and TM), while the magnitude of the humoral response to FIV p24 Gag was relatively attenuated (Figure 8). The LTNP cat (187) had lower titers directed against Gag p24 and Env (SU) than the FIV progressors (165, 184, and 186) and was most similar to the uninfected control animal 183.

CD134 (OX40) serves as the primary cell entry receptor for FIV [27,28]. The LTNP cat 187 did not demonstrate elevated anti-CD134 titers relative to the progressor cats 184, 165, and 186 (Figure 8). The animal with the highest anti-CD134 titers, cat 184, was classified as an FIV progressor based on progressive and profound CD4 T cell depletion (Figure 2). The magnitude of the anti-CD134 response in all of the other FIV-infected cats (including the LTNP) was more than 10-fold lower (with the exception of the initial measurement at 1 year PI for progressor cat 165).

Immunoglobulin isotype-specific ELISAs and plasma samples obtained throughout the FIV infection time course (1–12 years PI) indicate that the feline anti-FIV response is largely IgG-based (IgG FDG1-2 and IgG kappa) and not IgA or IgM (Figure 9). The anti-FIV IgG titers for LTNP cat 187 are similar to those for the progressor cats 165, 184, and 186. Interestingly, progressor cat 184 demonstrated the highest anti-FIV IgG immunoglobulin titers, anti-p24 Gag, anti-Env (SU), anti-Env (TM), and anti-CD134, relative to all the other FIV-infected cats.

### 3.6. Proviral DNA and RNA Isolated from Mesenteric Lymph Nodes

Proviral DNA was concurrently PCR amplified and quantified from the mesenteric LN obtained during the necropsies of all of the FIV-infected cats and ranged from 1.4 × 10^2^ (cat 186) to 2.0 × 10^4^ (cat 165) copies viral DNA/ million copies of feline GAPDH DNA. Viral RNA was amplified from LN of cat 165 (9.7 × 10^3^ copies viral RNA/ million copies GAPDH RNA) and was not detected (ND) in LN samples from cats 184, 187, and 186 (Figure 10). The positive and negative control reactions run in parallel were appropriate. The result for cat 165 is similar to the previously published results for viral DNA and RNA isolated from the mesenteric LN [21].

### 3.7. Experimentally FIV-Infected Cats Eventually Succumbed to Either Lymphoma or Chronic Renal Failure

For each animal, age at euthanasia, reason for euthanasia, and gross and microscopic lesions identified during the postmortem examinations are listed in Table 1, and images of select lesions are depicted in Figure 11, Figure 12 and Figure 13. Two of the FIV progressor cats (165 and 184) succumbed to multicentric lymphoma, while three of the cats (FIV progressor 186, FIV LTNP 187, and control 183) were euthanized due to complications of CRF. Control cat 185 was adopted out of the colony after cat 187 was euthanized in January 2021. Cat 185 was eventually euthanized at the owner’s request in November 2022, approximately 1.8 years later, for severe respiratory distress of undetermined etiology. A necropsy was not performed for cat 185. Although three of the cats were euthanized due to CRF (Figure 11 and Figure 13), all of the examined animals had some microscopic evidence of renal inflammation and renal tubular loss (interstitial nephritis). Although FIV progressor cats 165 and 184 had some histological evidence of interstitial nephritis, serum chemistry screens performed on the day of euthanasia demonstrated no evidence of azotemia for cat 165, while cat 184 had a slightly elevated BUN (34.6 mg/dL; normal < 33 mg/dL). Serum creatinine and phosphorus values were determined to be within normal limits for cat 184. In addition, the hematocrit and urine specific gravity values at the time of necropsy for cat 184 were 42.7% and >1.035, respectively, and were values inconsistent with CRF.

Peritonsillar ulcerative stomatitis in progressor cat 186 may be an oral manifestation of CRF-associated uremia. Four of the cats had gross and histological evidence of parathyroid gland hyperplasia (183, 184, 186, 187), likely due to renal secondary hyperparathyroidism. Thyroid hormone analysis (T4) was not a component of the serum chemistry screen; as a result, it was not determined if cat 165 had hyperthyroidism.

FIV progressor cat 184 had evidence of lymphoma in the myocardium of the heart, kidneys, intercostal musculature, diaphragm, and epidural space of the lumbar spinal canal (Figure 11 and Figure 12). The extradural spinal lymphoma lesion in cat 184 resulted in the clinical manifestation of acute paraparesis (dragging of hind limbs) through focal compression and degeneration of the white matter tracts of the affected segment of the spinal cord. Postmortem lesions in progressor cat 165 included independently arising T cell lymphomas of the cervical and mandibular lymph nodes and gut mucosa (described previously [21]). Lymphoid hyperplasia was identified grossly and microscopically in the lymph nodes (LTNP cat 187) and the splenic white pulp (progressor cat 186), and focally in the external inguinal lymph node of control cat 183 (Figure 12).

## 4. Discussion

In progressor animals, experimental infection with FIV lentivirus resulted in a progressive and profound immunodeficiency characterized by an initial loss of peripheral CD4 T cells [7], eventually followed by loss of both CD8 T cell and CD21 lymphocyte subsets during the mid to late asymptomatic phase. Although the U.S. Center for Disease Control’s definition of Stage 3 AIDS in human patients (<200 CD4 lymphocytes/µL blood; CDC.gov) is satisfied in FIV progressor cats, these cats nevertheless enter a prolonged asymptomatic phase featuring minimal to no clinical morbidity. During the late asymptomatic phase, the three progressor cats in this study, 165, 184, and 186, all had average CD4 counts of less than 200 cells/µL blood for 200 or more weeks (~4 years) without clinical evidence of substantive morbidity.

There is an apparent disconnect between humans infected with HIV and cats infected with FIV. Although both lentiviruses are associated with comparable lymphocyte depletion in their mammalian hosts, for undetermined reasons, the clinical effect of profound CD4 depletion on the feline host’s overall health is much attenuated relative to human patients. As a result, the asymptomatic phase is prolonged, and the terminal stage of FIV infection is relatively short. It is notable that this cohort of experimentally FIV-infected cats was housed and maintained in a controlled environment and protected from exposure to other animals at the UC Davis Feline Research Laboratory, possibly explaining the lack of evidence of terminal opportunistic infections. Interestingly, early FIV studies indicated that the terminal AIDS-like phase of the illness was mainly identified in naturally infected cats [16].

Three of the cats in the study reported here were euthanized due to CRF-associated morbidity. CRF, often manifesting morphologically as renal atrophy and tubulointerstitial nephritis, is common in aged cats, and most client-owned domestic cats older than 10 years of age have evidence of some degree of renal inflammation and parenchymal loss (personal observation, BGM). CRF results in metabolic dysfunction characterized by azotemia (elevated blood urea nitrogen (BUN), creatinine, and phosphorus), anorexia, various electrolyte abnormalities, dehydration, and non-regenerative anemia due to inadequate renal erythropoietin production. The cause of CRF in domestic cats is controversial, incompletely understood, and likely multifactorial [29]. A causal etiologic role for FIV in feline CRF remains controversial [30]. In one study, cats less than 11 years of age with CRF were significantly more likely to have serum antibodies against FIV than cats without CKD [31]. Types of FIV-associated renal disease that have been described include proteinuria (possibly as a result of immune complex deposition and glomerular injury) and, less often, azotemia and low urine specific gravity [32,33,34,35]. However, other studies have determined that the incidence of CRF is similar for both FIV-positive and FIV-negative cats [29]. It is important to note that although two of the FIV-infected cats in our study were diagnosed with CRF, one of the uninfected control animals also had CRF.

Lymphoma (lymphosarcoma) is a common outcome for FIV-infected cats [5,36]. The mechanism of lymphomagenesis in FIV-infected cats is controversial, and both direct (insertional mutagenesis) and indirect mechanisms (impaired immunosurveillance) have been proposed. In a study examining the rapid occurrence of lymphoma in four experimentally FIV-infected cats, the investigators reached the conclusion that neoplastic pathogenesis was secondary to FIV immune dysregulation [36]. Our research group reached the same conclusion of an indirect role for FIV in the pathogenesis of lymphoma for progressor cat 165 [21]. Although lymphoid depletion and involution (atrophy) of lymphoid tissues are consistent findings in tissues harvested from cats during the later stages of FIV infection [5], lymphoid hyperplasia was identified in the FIV-infected (187 and 186) and uninfected cats (183). Lymphoid hyperplasia has been described as a feature of acute FIV infection [5,36,37] and has also been recognized as a response to antigen stimulation in other immunosuppressive lentiviral infections like HIV and SIV [38,39]. In an earlier surgical biopsy study of this cohort performed at 5.5 years PI, all of the FIV-infected cats had marked follicular hyperplasia of the popliteal LN characterized by well-developed germinal centers with a prominent mantle cell rim and moderate paracortical atrophy [40]. Although islet amyloidosis can be associated with diabetes mellitus, there was no clinical evidence of this endocrinopathy (hyperglycemia) in the affected animals.

Studies have indicated that antibodies directed against CD134 have been associated with improved health and survival as these antibodies have been shown to indirectly block FIV entry ex vivo, and the presence of antibodies to CD134 has been associated with lower viral loads [41]. As a result, we were interested in determining the anti-CD134 status in our FIV-infected cohort. Unexpectedly, we found that the LTNP cat 187 did not have elevated CD134 titers relative to the progressors and that cat 184 had the highest titers against CD134. Interestingly, profound CD8 lymphocyte depletion in cat 184 340–425 weeks PI eventually recovered to the normal range (500–2500 cells/µL blood) from 450 weeks PI onward. Whether there is any association between this unusual CD8 T cell rebound and elevated anti-CD134 titers was not determined. Although progressor cat 184 had gross and histological lesions at the time of necropsy that were similar to progressor cat 165 (multicentric lymphoma, islet amyloidosis, interstitial nephritis), cat 184 lived for 3.8 years longer than cat 165. It is not clear if the difference in relative longevity between 184 and 165 might be related to the peripheral CD8 T cell rebound or differences in serologic response between 184 and 165.

HIV-infected controllers and long-term non-progressor patients have been extensively studied in human medicine [42,43]. To our knowledge, cat 187 is one of the few, or perhaps only, experimentally FIV-infected animal that has had consistent, long-term documentation supportive of LTNP status. We have previously defined an FIV LTNP as a persistently asymptomatic animal that maintains a peripheral blood CD4 T cell count indistinguishable from FIV-negative control cats [40]. Our prior investigations have indicated that, relative to the FIV progressors, cat 187 had lower proviral loads in PBMC and tissue (LN), and viral RNA was detected at a lower copy number in LNs [40]. Although it was possible to reactivate replication competent virus from cat 187’s LN, viral reactivation took longer than observed for the three progressor cats [40]. In the data presented herein, LTNP 187 showed lower immunoglobulin titers directed against Gag p24 and Env (SU) than the FIV progressor cats and did not exhibit elevated CD134 titers (“protective antibodies”) relative to the FIV progressors. In addition, there was no distinct genetic signature in the amplified proviral LTR or *gag* gene (e.g., large deletion or rearrangement), clearly delineating the LTNP from the progressor animals in the tissue and PBMC-derived sequences. The inoculating viral swarm was the same for all the infected animals [7].

The mechanisms conferring LTNP status have been extensively studied for HIV patients and appear to involve multiple host factors: production/destruction of T cells, IL-7/IL-7R, status of lymphoid architecture, balance of pro-/anti-inflammatory cells, lentivirus-specific immune response, size of the viral reservoir [43], and host MHC genotype [44]. Viral genotype affecting replication fitness can also play a role in host control of viral replication [44]. Extensive data unambiguously show that HIV-1 controllers are immunologically different from progressors in the production, destruction, and regulation of CD4 T cells [43]. The mechanism(s) governing LTNP status in FIV-infected cats may involve similar host factors but requires further research.

Cats experimentally infected with FIV have a profound and progressive loss of multiple peripheral leukocytes. Data from HIV-infected humans suggest that loss of critical lymphocyte subsets, such as CD4 T cells, results in an impaired adaptive immune system [45]. Whether these immunologic effects of chronic lentiviral infection in cats are causally related, either directly or indirectly, to the terminal manifestation of lymphoma and/or CRF is difficult to conclusively prove.

## Figures and Tables

**Figure 1 viruses-15-01775-f001:**
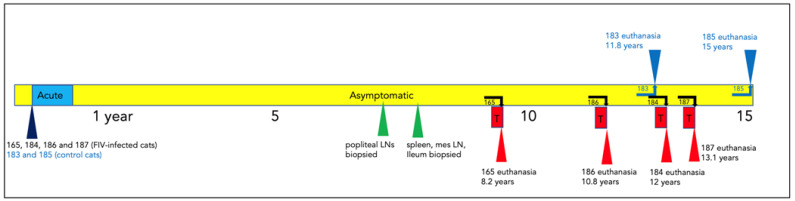
Clinical timeline for all of the study animals. The acute (blue box), asymptomatic (yellow), and terminal (red) phases of FIV infection are depicted for each animal. Survival surgical biopsy procedures are depicted by green triangles. The age of each animal at the time of euthanasia is indicated (blue and red triangles). The term “mes” denotes mesenteric.

**Figure 2 viruses-15-01775-f002:**
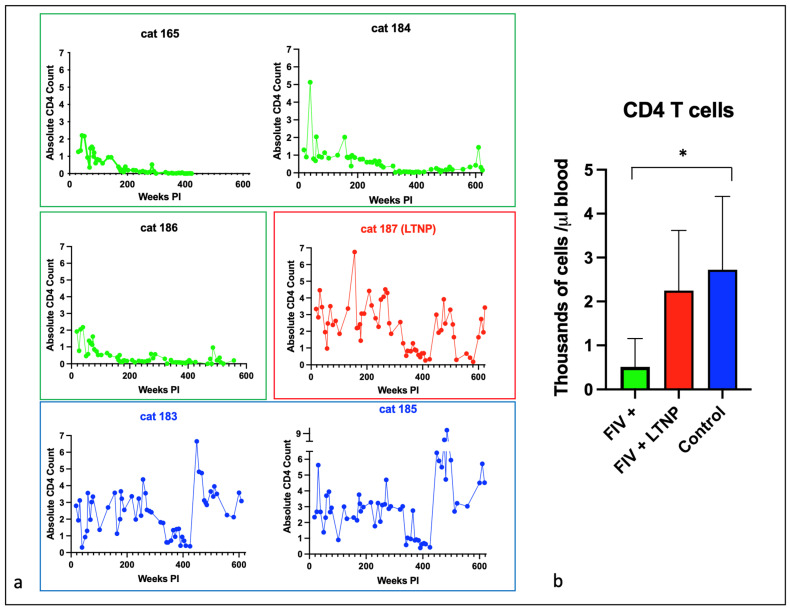
Peripheral CD4 T cells decline over time in FIV progressor cats but not in non-progressor and control animals. (**a**) The absolute number of peripheral CD4 T cells in progressor cats (165, 184, 186—green), long term non-progressor (LTNP, 187, red), and uninfected control cats (183 and 185, blue) are depicted vs. time post-infection (PI). (**b**) The aggregated average of all CD4 lymphocyte data is depicted as a colored bar with standard deviations for progressor (FIV+, green), LTNP (red), and control animals (blue). An asterisk indicates a significant difference of means (*, *p* < 0.05).

**Figure 3 viruses-15-01775-f003:**
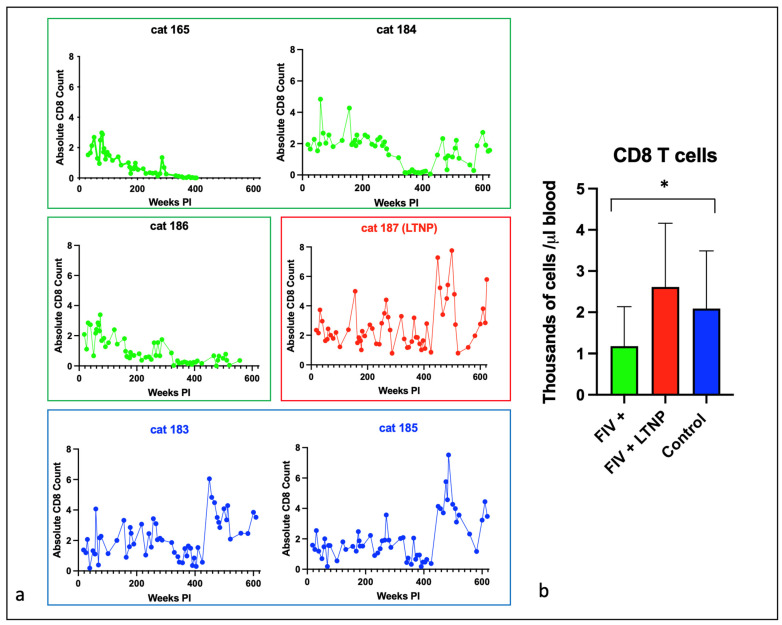
Peripheral CD8 T cells decline over time in most FIV progressor cats but not in non-progressor and control animals. (**a**) The absolute number of peripheral CD8 T cells in progressor cats (165, 184, 186—green), long term non-progressor (LTNP, 187, red), and uninfected control cats (183 and 185, blue) are depicted vs. time post-infection (PI). (**b**) The aggregated average of all CD8 T lymphocyte data is depicted as a colored bar with standard deviations for progressor (FIV+, green), LTNP (red), and control animals (blue). An asterisk indicates a significant difference of means (*, *p* < 0.05).

**Figure 4 viruses-15-01775-f004:**
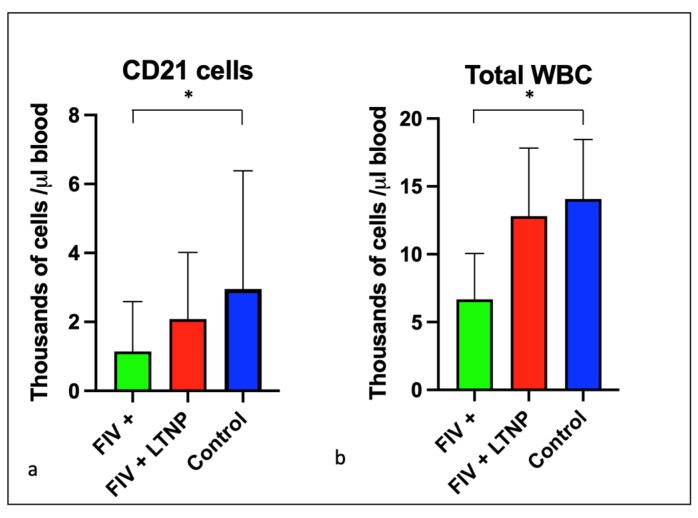
CD21 cells and total white blood cells are reduced in progressor cats relative to non-progressor and control animals. The aggregated average and standard deviations of (**a**) peripheral CD21 cells (**b**) and total WBC for progressor cats (FIV+, green), long term non-progressor (red), and control animals (blue). An asterisk indicates a significant difference of means (*, *p* < 0.05).

**Figure 5 viruses-15-01775-f005:**
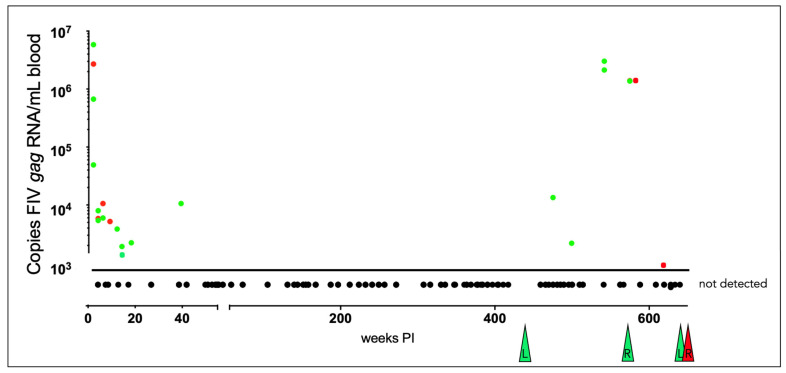
Viral gag RNA is intermittently detectable in the blood of the FIV-infected cats during the acute and late asymptomatic phases of infection. Real-time RT-PCR data from individual cats are indicated by color: green (progressors 165, 184, 186) and red (LTNP 187). Samples are plotted as copies of FIV *gag* RNA/mL blood vs. the number of weeks post-infection (PI). Samples below the limit of detection (not detected, <800 copies RNA/mL blood) are plotted below the x-axis, and colored triangles indicate the time of euthanasia for each FIV-infected animal. “L” indicates lymphoma, while “R” indicates renal failure as a cause of euthanasia.

**Figure 6 viruses-15-01775-f006:**
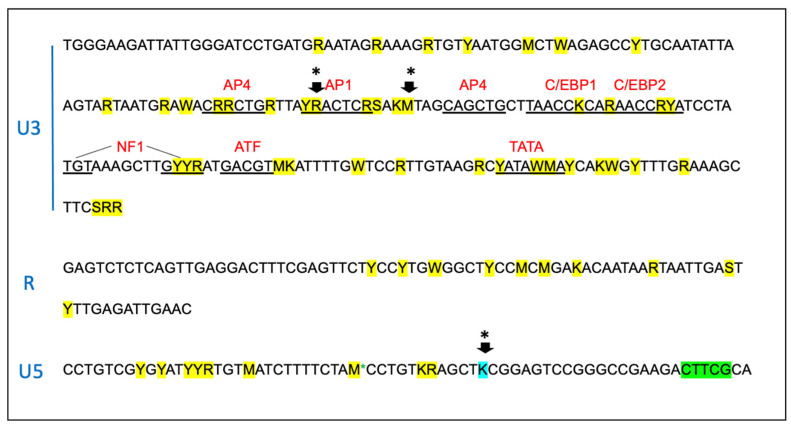
The sequence of the proviral promoter amplified from the tissues of the FIV-infected cats is unstable over time. The sequence of the entire long terminal repeat (LTR) is depicted for the inoculating virus. SNPs identified within the amplified and cloned sequences of tissue-derived provirus are indicated by yellow highlights. Green indicates a block of deleted nucleotides in the U5 region, and blue indicates an SNP in the inoculating viral sequence. Short arrows with an asterisk (*) represent common SNPs isolated from multiple cats at multiple time points. The standard IUPAC nucleotide code is used. Recognized transcription factor binding motifs are indicated in red.

**Figure 7 viruses-15-01775-f007:**
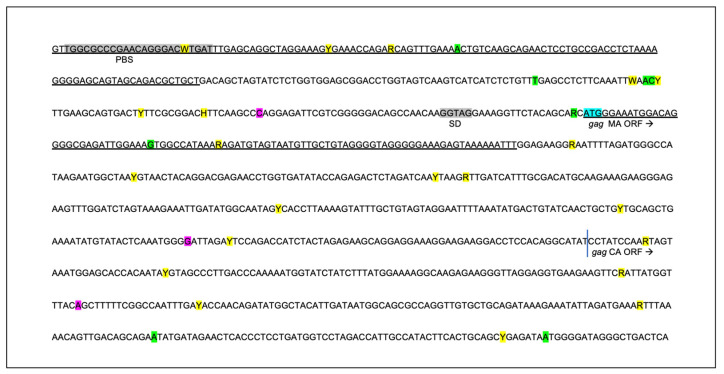
The sequence of the 5′ aspect of the proviral gag region amplified from the tissues of the FIV-infected cats is unstable over time. Approximately 1000 nucleotides of the FIV *gag* gene encompassing the primer binding site (PBS), *gag matrix* (MA), and *gag capsid* (CA) are indicated. Sequences known to be important for efficient encapsidation are underlined, and the splice donor site (SD) is indicated in grey. SNPs are indicated by yellow highlight, deletions by green, and insertions are indicated by purple. The initiating ATG codon for *gag matrix* (MA ORF) is indicated by a blue highlight; the vertical blue line is the beginning of *gag capsid* (CA ORF). The standard IUPAC nucleotide code is used.

**Figure 8 viruses-15-01775-f008:**
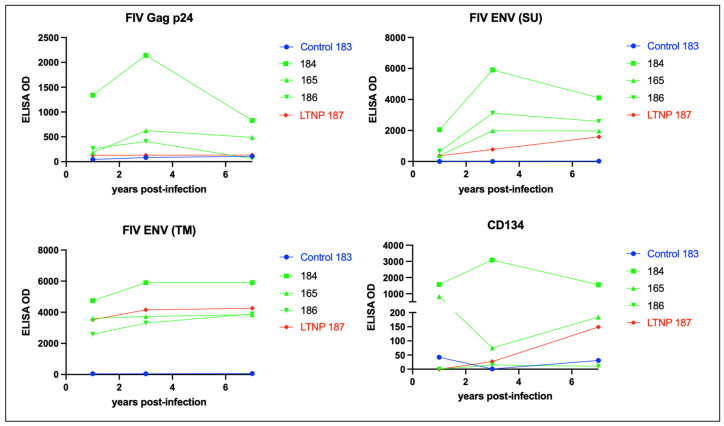
FIV-infected cats mount a strong humoral response against Env TM and a variably strong response to Env SU, Gag p24, and CD134. ELISA titers are plotted against time post-infection for the FIV progressors (green), non-progressor (red), and uninfected control (blue). The non-progressor cat (LTNP 187) has a high Env TM antibody titer and a relatively weak humoral response against Env-SU, Gag p24, and the primary FIV receptor CD134.

**Figure 9 viruses-15-01775-f009:**
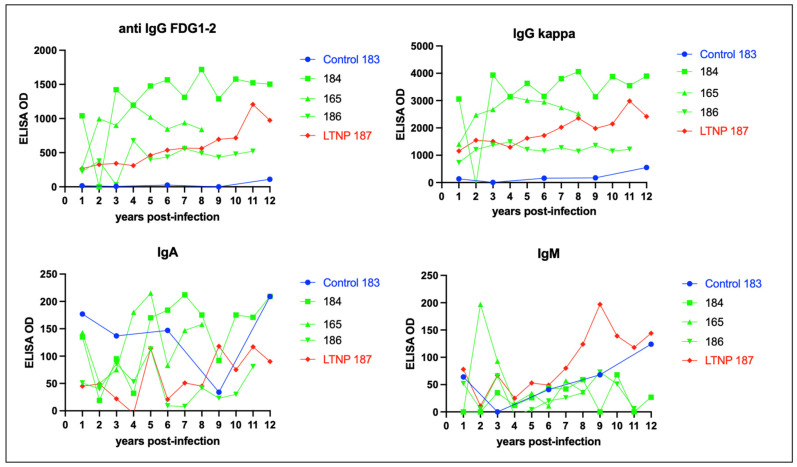
The isotype of the humoral response against FIV is IgG, not IgA or IgM. ELISA titers for anti-FIV IgG (FDG1-2 and kappa), IgA, and IgM are plotted against time post-infection for the FIV progressors (green), non-progressor (red), and uninfected control (blue).

**Figure 10 viruses-15-01775-f010:**
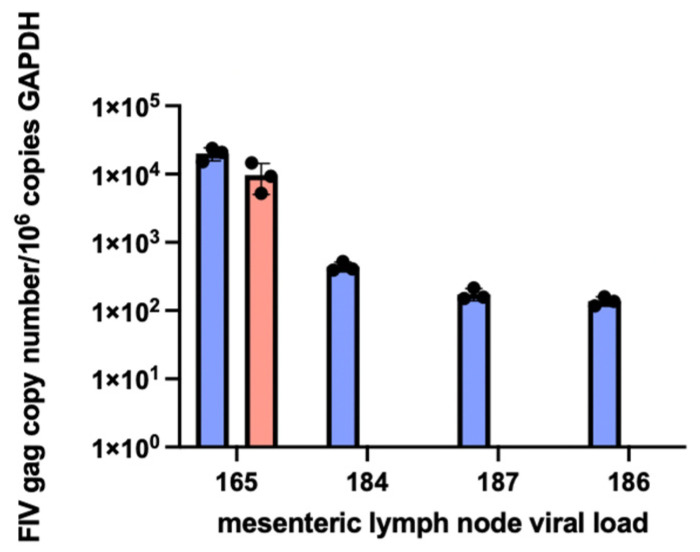
Proviral DNA (blue bars) is detected in the mesenteric lymph node tissue in all of the FIV-infected cats, while viral RNA (cDNA, pink bar) is only detectable in the LN tissue of progressor cat 165. Mesenteric LN was harvested at the time of necropsy. Real-time PCR data are normalized to copies of host feline GAPDH.

**Figure 11 viruses-15-01775-f011:**
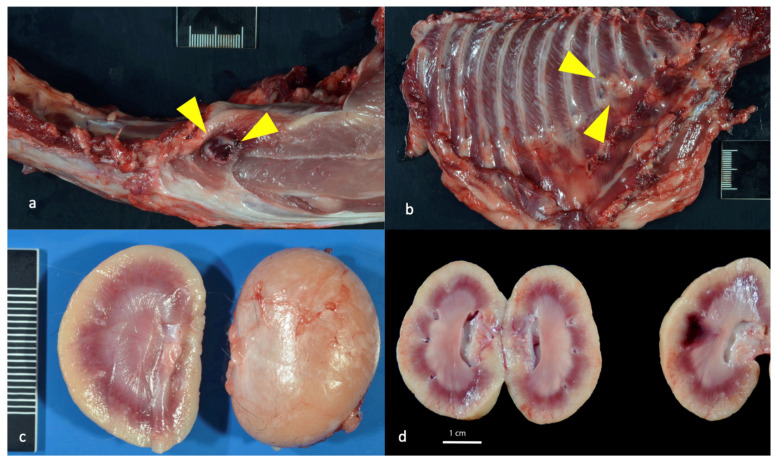
Gross lesions identified at the time of necropsy include lymphoma and renal atrophy. (**a**,**b**) Discrete, purple-to-tan neoplastic nodules (lymphoma) identified within the diaphragm and intercostal musculature of progressor cat 184 are depicted by yellow triangles. (**c**) Bilateral kidneys of control cat 183 were small, firm, and rounded. The right kidney is 6.8 g and is 3 × 2.7 × 2.5 cm (renal atrophy). (**d**) In progressor cat 186, bilateral kidneys are moderately atrophied with an undulating cortical surface. Bilateral kidneys have multiple, small (less than 1 mm) diameter cyst-like structures in the outer cortical parenchyma. and the right kidney has a pale, bulging outer cortex and a dark purple inner cortex (acute infarct).

**Figure 12 viruses-15-01775-f012:**
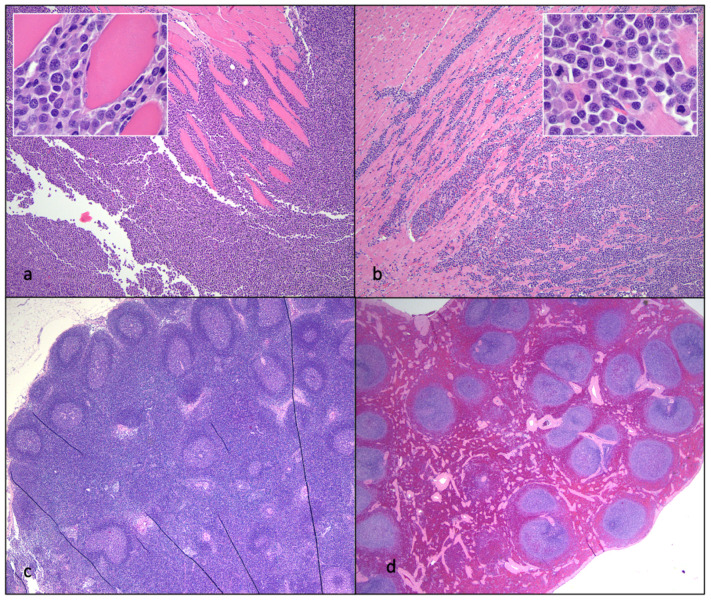
Histological features of lymphoma and lymphoid hyperplasia in FIV-infected cats. (**a**,**b**) Disruption of the intercostal (**a**) and cardiac (**b**) musculature by sheets of neoplastic lymphocytes is evident in tissues from progressor 184 (lymphoma). Insets—Neoplastic lymphoblasts (blue cells) infiltrate and isolate skeletal myocytes and cardiac myocytes (pink cells). (**c**) The mesenteric lymph node has numerous well-organized cortical follicles with a germinal center and peripheral mantle zone (lymphoid hyperplasia, non-progressor cat 187). (**d**) The splenic white pulp has numerous hyperplastic lymphoid follicles (lymphoid hyperplasia, progressor 186).

**Figure 13 viruses-15-01775-f013:**
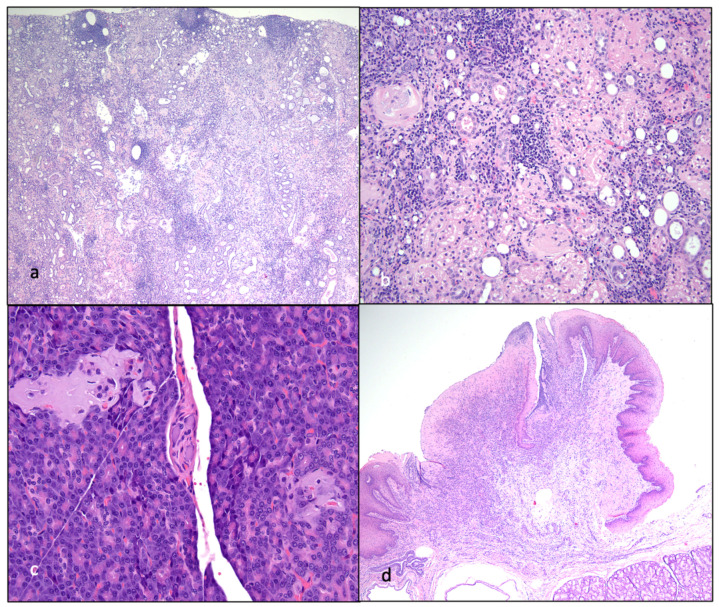
Histological features of nephritis, islet amyloidosis, and ulcerative stomatitis in FIV-infected cats. (**a**) The renal cortex has a loss of tubules and numerous interstitial aggregates of lymphocytes and plasma cells (interstitial nephritis, progressor cat 186). (**b**) The renal interstitium is multifocally infiltrated by lymphocytes and plasma cells, and scattered glomeruli are sclerotic (interstitial nephritis, non-progressor cat 187). (**c**) Endocrine cells within the pancreatic Islets of Langerhans are largely replaced by amphophilic amyloid material (islet amyloidosis, non-progressor cat 187). (**d**) The oral mucosa adjacent to the tonsil and diffuse salivary gland tissue is necrotic and ulcerated, and the lamina propria is infiltrated by large numbers of lymphocytes and plasma cells (peritonsillar ulcerative stomatitis, progressor cat 186).

**Table 1 viruses-15-01775-t001:** Terminal clinical abnormalities and pathologic findings.

	Age at Euthanasia	Reason for Euthanasia	Major Gross Lesions	Major Histological Lesions
**165 FIV progressor**	8.2 years	Anorexia, weight loss, fever, pancytopenia	Essentially none	Lymphoma (LN and SI) LN atrophy (mesenteric) Islet amyloidosis Thyroid gland adenomatous hyperplasia Interstitial nephritis
**186 FIV progressor**	10.8 years	CRF: azotemia, hypercalcemia, hyperphosphatemia, NR anemia	Renal atrophy Parathyroid gland hyperplasia Splenic lymphoid hyperplasia	Interstitial nephritis Lymphoid hyperplasia (spleen and LN) Parathyroid gland hyperplasia Peritonsillar ulcerative stomatitis
**184 FIV** **progressor**	12 years	Paraparesis (12 h)	Multicentric lymphoma Tooth resorption and crown fractures	Lymphoma (spinal canal, heart, kidneys, skeletal muscle, LN) Islet amyloidosis Interstitial nephritis Parathyroid gland hyperplasia
**187 FIV LTNP**	13.1 years	CRF: azotemia, dehydration, isosthenuria, NR anemia, weight loss	Renal atrophy Generalized lymphadenomegaly Parathyroid gland hyperplasia Tooth resorption and crown fractures	Interstitial nephritis Lymphoid hyperplasia Parathyroid gland hyperplasia Islet amyloidosis Enteritis
**183 control**	11.8 years	CRF: azotemia, dehydration, isosthenuria, NR anemia, weight loss	Renal atrophy Parathyroid gland hyperplasia Focal external inguinal LN hyperplasia	Interstitial nephritis Parathyroid gland hyperplasia Enteritis Cortical lymphoid hyperplasia
**185 control**	15 years	Respiratory distress	ND	ND

Green background denotes FIV progressors, red FIV non-progressor, blue control animals. CRF—chronic renal failure, LN—lymph node, SI—small intestine, ND—not determined.

## Data Availability

Data is contained within the article.

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
