# Peer review of "The Late Asymptomatic and Terminal Immunodeficiency Phases in Experimentally FIV-Infected Cats—A Long-Term Study"

_viruses, 2023, doi:10.3390/v15081775_

Round 1

Reviewer 1 Report

Line 47: I’m not sure that we are currently using this terminology, it tends to confuse and frighten pet owners. These are old references.

Section 3.1: I understand that this is a virus journal, I do wonder if placing Table 1 sooner would help with context, particularly the causes of death as that is a primary focus of the manuscript. If the other publications don’t have the details, these clinical findings by individual cats would be helpful for practicing veterinarians to see if there are any tendencies for the blood work from renal vs lymphoma cats to differ.

Line 235: looks like it may be that progressor cat 184 the rebound led to longer life? Thoughts?

Love that the figure starts with the take home message of the findings!

Fig 2: low power to detect differences and if this graph 2b is the mean of all counts, that isn’t a t-test; t-tests are for independent samples where the mean and standard deviation of all the different independent samples (usually from different animals) are compared. And because this is a comparison of 3 groups it would be ANOVA. Or two t-tests with a Bonferroni or other multiple comparison correction. However, the patterns are clearly different per the graphs, so the conclusions are correct.

Figure 3: not sure I believe that there are significant differences among the 3 progressor cats and the 1 LTNP and 2 control cats. That is likely due to an erroneous application of the t-test. And the graphs of cat 184 and control cats aren’t that different.  Hard to tell with the sample size and lack of clarity about the summary stats.  I think that the b green graphics are the average lifetime counts of the 3 FIV+ cats but not sure what the SD really is. If it is the variability/SD of only the 3 averages of the cats, then that is correct. But it would seem not because there is an error bar for the LTNP single cat.

Similar concern for Figure 4 and the conclusions about significant differences. I think that the pattern for FIV + vs LTNP/Control are clear but with only a single LTNP cat, ultimately, for all figures, no statistical comparison can be made because there is no between-cat variability in that group. Therefore, the only valid conclusion is that the FIV+ cats appear to have lower mean counts than the other 3 cats (based on these data which I don’t know are accurate for the analysis I’m talking about). And the only statistical comparison would be between the FIV+ and the Control cats using 1 mean per cat and the SD of those 3 means (not for all counts) for each group. So, the italic results should be adjusted (unless reanalysis shows similar findings to the current analysis) to indicate that this is a trend or pattern.

Figure 5: the LTNP cat is red in the other figures; please continue that here for ease of understanding and comparison. And the summary statement in italics isn’t quite accurate because RNA from cat 165 was only detected in the acute stage.

Line 341: but not any clinical distinctions or necropsy/histopathology differences? Please add or clarify.

Figure 9: please indicate why the second control cat is not in the graphs.

Table 1: it appears from the highlighted text that 165 and 184 both also had some level of renal disease. What did the clinical pathology show? Was it normal or did it support renal disease that was just not severe enough to be life threatening or caught by blood work?  And for 165, was the thyroid abnormality apparent by laboratory work? Was this a functional tumor in which case some of the weight loss could be related to that? Please clarify in text for clinical relevance.

Lines 384-6: this is discussion. And the findings for these cats should be included here.

Line 400, first sentence. Also a discussion topic. Also most of lines 416-24. Please just include the findings here and then discuss the implications and references in the discussion section. Same for following paragraph…where references are included, please move to discussion section.

Figure 13 and following text. Any additional information about the nephritis to put in the results and then include in the discussion section?

Line 460-2: but what about free roaming cats or cats who live with other animals, or many different humans? Many of these cats also live a long time. These statements need to be toned down; the acknowledgement of differences in lifestyles is good and important but may not be the reason, particularly with only 3 cats as data points.

Author Response

viruses-2557308

Reviewer 1

Line 47: I’m not sure that we are currently using this terminology, it tends to confuse and frighten pet owners. These are old references.

This sentence, and all references to AIDS or FAIDS, have been deleted.

Section 3.1: I understand that this is a virus journal, I do wonder if placing Table 1 sooner would help with context, particularly the causes of death as that is a primary focus of the manuscript. If the other publications don’t have the details, these clinical findings by individual cats would be helpful for practicing veterinarians to see if there are any tendencies for the blood work from renal vs lymphoma cats to differ.

Table 1 has been moved to the beginning of the Results section, directly after Figure 1.

Line 235: looks like it may be that progressor cat 184 the rebound led to longer life? Thoughts?

The following text has been added to the manuscript at this line:

It should be noted that cat 184 lived longer than the other two progressor animals- 3.8 years longer than cat 165 and 1.2 years longer than cat 186.  Whether the peripheral CD8 T cell rebound had any role in this discrepancy in longevity remains speculative.

Love that the figure starts with the take home message of the findings!

Fig 2: low power to detect differences and if this graph 2b is the mean of all counts, that isn’t a t-test; t-tests are for independent samples where the mean and standard deviation of all the different independent samples (usually from different animals) are compared. And because this is a comparison of 3 groups it would be ANOVA. Or two t-tests with a Bonferroni or other multiple comparison correction. However, the patterns are clearly different per the graphs, so the conclusions are correct.

This error of statistical interpretation has been corrected in the figures (2-4), Figure Legends, and text.  T tests have now been performed to compare the means of only the 3 progressor animals vs. the 2 control animals, and not against the single LTNP cat.

Figure 3: not sure I believe that there are significant differences among the 3 progressor cats and the 1 LTNP and 2 control cats. That is likely due to an erroneous application of the t-test. And the graphs of cat 184 and control cats aren’t that different.  Hard to tell with the sample size and lack of clarity about the summary stats.  I think that the b green graphics are the average lifetime counts of the 3 FIV+ cats but not sure what the SD really is. If it is the variability/SD of only the 3 averages of the cats, then that is correct. But it would seem not because there is an error bar for the LTNP single cat.

See statement above concerning the t tests for Figures 2-4.

Similar concern for Figure 4 and the conclusions about significant differences. I think that the pattern for FIV + vs LTNP/Control are clear but with only a single LTNP cat, ultimately, for all figures, no statistical comparison can be made because there is no between-cat variability in that group. Therefore, the only valid conclusion is that the FIV+ cats appear to have lower mean counts than the other 3 cats (based on these data which I don’t know are accurate for the analysis I’m talking about). And the only statistical comparison would be between the FIV+ and the Control cats using 1 mean per cat and the SD of those 3 means (not for all counts) for each group. So, the italic results should be adjusted (unless reanalysis shows similar findings to the current analysis) to indicate that this is a trend or pattern.

See statement above concerning t tests for Figures 2-4.

Figure 5: the LTNP cat is red in the other figures; please continue that here for ease of understanding and comparison. And the summary statement in italics isn’t quite accurate because RNA from cat 165 was only detected in the acute stage.

Figure 5 has been altered such that the data for the progressors are green and the nonprogressors is red.  The Figure Title has been altered by the word “intermittently”.

Line 341: but not any clinical distinctions or necropsy/histopathology differences? Please add or clarify.

Added to the Discussion section:

Although progressor cat 184 had gross and histological lesions at the time of necropsy that were similar to progressor cat 165 (multicentric lymphoma, islet amyloidosis, interstitial nephritis), cat 184 lived for 3.8 years longer than 165.  It is not clear if the difference in relative longevity between 184 and 165 is related to the peripheral CD8 T cell rebound or to differences in serologic response between 184 and 165.

Figure 9: please indicate why the second control cat is not in the graphs.

The ELISA assays were performed with serial samples derived from a single control cat due to limited available space on a 96 well plate.

Table 1: it appears from the highlighted text that 165 and 184 both also had some level of renal disease. What did the clinical pathology show? Was it normal or did it support renal disease that was just not severe enough to be life threatening or caught by blood work?  And for 165, was the thyroid abnormality apparent by laboratory work? Was this a functional tumor in which case some of the weight loss could be related to that? Please clarify in text for clinical relevance.

Renal lesions in cats 184 and 165 were likely of insufficient severity to manifest as CRF. The following has been added to the Discussion section:

Although progressor cats 165 and 184 had histological evidence of interstitial nephritis, serum chemistry screens performed on the day of euthanasia demonstrated no evidence of azotemia for cat 165 while cat 184 had a slightly elevated BUN (34.6 mg/dl, normal < 33 mg/dl) and creatinine and phosphorus were considered to be within normal limits.

And Results section (3.7):

Thyroid hormone analysis (T4) was not a component of the serum chemistry screen; as a result, it was not determined if cat 165 had hyperthyroidism.

Lines 384-6: this is discussion. And the findings for these cats should be included here.

These lines have been moved to the Discussion section.

Line 400, first sentence. Also a discussion topic. Also most of lines 416-24. Please just include the findings here and then discuss the implications and references in the discussion section. Same for following paragraph…where references are included, please move to discussion section.

These lines have been moved to the Discussion section.

Figure 13 and following text. Any additional information about the nephritis to put in the results and then include in the discussion section?

More information on nephritis and a possible relationship between nephritis and FIV have been added to the Discussion.

Line 460-2: but what about free roaming cats or cats who live with other animals, or many different humans? Many of these cats also live a long time. These statements need to be toned down; the acknowledgement of differences in lifestyles is good and important but may not be the reason, particularly with only 3 cats as data points.

This line has been deleted.

Reviewer 2 Report

I thought this was an excellent paper. Its major limitation is that it involves only 4 cats. But these long-term experiments are VERY expensive. I don’t have many issues with the raw data. But I do have some issues:

1.       NAV calls the mandibular lymph nodes, NOT submandibular

2.       Please describe the  immunophenotype of the lymphoma that occurred in the cats.

3.       There is good discussion of how FIV might contribute to  lymphomagenesis. But there is strong evidence from the 3 cats that FIV contributes to kidney disease. The authors COMPLETELY gloss over this point. Yet there is a lot of work – going back to the Poli papers, that look at the association between long standing FIV and CKD in cats

See for example:

Association between naturally occurring chronic kidney disease and feline immunodeficiency virus infection status in cats

JD White, R MalikJM Norris… - Journal of the American …, 2010 - Am Vet Med Assoc

… FIV gp40 than were cats without CKD. It cannot be definitively established from results of this
study … infection with 
FIV preceded the development of CKD, and the role, if any, of FIV in the …

Save Cite Cited by 25 Related articles All 8 versions

[PDF] wiley.comFull View

Risk factors for development of chronic kidney disease in cats

NC Finch, HM SymeJ Elliott - Journal of Veterinary Internal …, 2016 - Wiley Online Library

… cats with surgically induced models of kidney disease did not identify effects of dietary protein
content on renal function.37, 38 Senior diets for 
cats … not routinely screened for FIV status. It …

Save Cite Cited by 125 Related articles All 17 versions

Infectious Agents in Feline Chronic Kidney Disease: What Is the Evidence?

K Hartmann, MG Pennisi… - … in Small Animal …, 2020 - advancesinsmallanimalcare.com

… Chronic kidney disease (CKD) is among the most common diseases of cats and affects in
particular the elderly 
cat population. The overall estimated prevalence in a large study from the …

Save Cite Cited by 4 Related articles

[PDF] proquest.com

Clinical evaluation of dietary modification for treatment of spontaneous chronic kidney disease in cats

SJ Ross, CA Osborne, CA Kirk, SR Lowry… - Journal of the …, 2006 - Am Vet Med Assoc

… -term safety and effectiveness of renal diets in cats with naturally occurring kidney disease.
… the study was complete; 
cats with diabetes mellitus, hyperthyroidism, FeLV, or FIV infection; …

Save Cite Cited by 307 Related articles All 11 versions

[PDF] tci-thaijo.org

The risk factors of having infected feline leukemia virus or feline immunodeficiency virus for feline naturally occurring chronic kidney disease

K Piyarungsri, S Tangtrongsup… - Veterinary …, 2020 - he02.tci-thaijo.org

… virus (FIV) caused kidney problems. … kidney disease (CKD) and to estimate the possible
association between CKD and infection with either FeLV or 
FIV or with both FeLV and FIV in cats

Save Cite Cited by 2 Related articles All 4 versions 

[PDF] wiley.com

Chronic kidney disease in cats attending primary care practice in the UK: a VetCompassTM study

M Conroy, DC BrodbeltD O'NeillYM Chang… - Veterinary …, 2019 - Wiley Online Library

Chronic kidney disease (CKD) is a frequent diagnosis in cats attending primary care … death
in 
cats aged over five years, yet there is limited published research for CKD in cats attending …

Save Cite Cited by 41 Related articles All 10 versions

[PDF] core.ac.uk

Current understanding of the pathogenesis of progressive chronic kidney disease in cats

RE Jepson - Veterinary Clinics: Small Animal Practice, 2016 - vetsmall.theclinics.com

… Chronic kidney disease (CKD) is a common condition identified in cats at both general
practice … The term CKD is used to imply alteration in structure or function of the 
kidney that has …

Save Cite Cited by 65 Related articles All 8 versions

[HTML] sagepub.comFull View

Chronic kidney disease in aged cats: clinical features, morphology, and proposed pathogeneses

CA Brown, J Elliott, CW Schmiedt… - Veterinary …, 2016 - journals.sagepub.com

… complex glomerulonephritis have been described in FIV-positive cats. In addition to these
… noted in 
FIV-positive cats. However, in comparing FIV-infected and FIV-noninfected cats in …

Save Cite Cited by 196 Related articles All 11 versions

[PDF] researchgate.net

Risk factors associated with the development of chronic kidney disease in cats evaluated at primary care veterinary hospitals

JP Greene, SL Lefebvre, M Wang… - Journal of the …, 2014 - Am Vet Med Assoc

… Another study 8 of the potential association between FIV infection and CKD in cats revealed
no association between the 2 
diseases but did find a significant association between FIV …

Save Cite Cited by 123 Related articles All 9 versions

[HTML] sagepub.comFull View

Risk and protective factors for cats with naturally occurring chronic kidney disease

K Piyarungsri… - Journal of feline medicine …, 2017 - journals.sagepub.com

… an association between infectious diseases and CKD in catsFIV infection has been found
… of renal tissues of 
cats experimentally and naturally infected with FIV showed renal changes, …

Save Cite Cited by 22 Related articles All 8 versions

So, from my way of thinking, the impact of long standing FIV is  a long erm decline in the CD4 count, not much in the way of immunosuppression related diseases (PCP, toxoplasmosis, cryptococcosis, MAC, etc) but the cats have a foreshortened life and die of CKD or lymphoma when they are about 10 years old. The problem – is that CKD and lymphoma are common diseases of old cats and common causes of death, so in many studies and for many deaths, unless FIV testing is routine, clinicians might miss this IMPORTANT association – because they just diagnose the lymphoma or CKD and MISS the underlying FIV infection. And this fits in what i have seen clinically. It would so good to see a prospective study looking at a much larger cohort kept in people houses (not a research facility) – I bet the strong association is between FIV and mortality, i.e. FIV positive cats don’t live as long as FIV negative cats, with earlier onset of the common diseases of the older cat (CKD and lymphoma).

Author Response

viruses-2557308

Reviewer 2

I thought this was an excellent paper. Its major limitation is that it involves only 4 cats. But these long-term experiments are VERY expensive. I don’t have many issues with the raw data. But I do have some issues:

  1. NAV calls the mandibular lymph nodes, NOT submandibular

This editing change has been made.

  1. Please describe the immunophenotype of the lymphoma that occurred in the cats.

Immunohistochemistry and T cell clonality testing determined that progressor cat 165 had independently arising T cell lymphomas in the peripheral lymph nodes and gut.   The immunophenotype of the lymphoma in progressor cat 184 was not determined.  Information has been added to the Results section to this effect.

  1. There is good discussion of how FIV might contribute to  lymphomagenesis. But there is strong evidence from the 3 cats that FIV contributes to kidney disease. The authors COMPLETELY gloss over this point. Yet there is a lot of work – going back to the Poli papers, that look at the association between long standing FIV and CKD in cats

See for example:

Association between naturally occurring chronic kidney disease and feline immunodeficiency virus infection status in cats

JD White, R Malik, JM Norris… - Journal of the American …, 2010 - Am Vet Med Assoc

… FIV gp40 than were cats without CKD. It cannot be definitively established from results of this
study … infection with FIV preceded the development of CKD, and the role, if any, of FIV in the …

Save Cite Cited by 25 Related articles All 8 versions

[PDF] wiley.comFull View

Risk factors for development of chronic kidney disease in cats

NC Finch, HM Syme, J Elliott - Journal of Veterinary Internal …, 2016 - Wiley Online Library

… cats with surgically induced models of kidney disease did not identify effects of dietary protein
content on renal function.37, 38 Senior diets for cats … not routinely screened for FIV status. It …

Save Cite Cited by 125 Related articles All 17 versions

Infectious Agents in Feline Chronic Kidney Disease: What Is the Evidence?

K Hartmann, MG Pennisi… - … in Small Animal …, 2020 - advancesinsmallanimalcare.com

… Chronic kidney disease (CKD) is among the most common diseases of cats and affects in
particular the elderly cat population. The overall estimated prevalence in a large study from the …

Save Cite Cited by 4 Related articles

[PDF] proquest.com

Clinical evaluation of dietary modification for treatment of spontaneous chronic kidney disease in cats

SJ Ross, CA Osborne, CA Kirk, SR Lowry… - Journal of the …, 2006 - Am Vet Med Assoc

… -term safety and effectiveness of renal diets in cats with naturally occurring kidney disease.
… the study was complete; cats with diabetes mellitus, hyperthyroidism, FeLV, or FIV infection; …

Save Cite Cited by 307 Related articles All 11 versions

[PDF] tci-thaijo.org

The risk factors of having infected feline leukemia virus or feline immunodeficiency virus for feline naturally occurring chronic kidney disease

K Piyarungsri, S Tangtrongsup… - Veterinary …, 2020 - he02.tci-thaijo.org

… virus (FIV) caused kidney problems. … kidney disease (CKD) and to estimate the possible
association between CKD and infection with either FeLV or FIV or with both FeLV and FIV in cats

Save Cite Cited by 2 Related articles All 4 versions 

[PDF] wiley.com

Chronic kidney disease in cats attending primary care practice in the UK: a VetCompassTM study

M Conroy, DC Brodbelt, D O'Neill, YM Chang… - Veterinary …, 2019 - Wiley Online Library

Chronic kidney disease (CKD) is a frequent diagnosis in cats attending primary care … death
in cats aged over five years, yet there is limited published research for CKD in cats attending …

Save Cite Cited by 41 Related articles All 10 versions

[PDF] core.ac.uk

Current understanding of the pathogenesis of progressive chronic kidney disease in cats

RE Jepson - Veterinary Clinics: Small Animal Practice, 2016 - vetsmall.theclinics.com

… Chronic kidney disease (CKD) is a common condition identified in cats at both general
practice … The term CKD is used to imply alteration in structure or function of the kidney that has …

Save Cite Cited by 65 Related articles All 8 versions

[HTML] sagepub.comFull View

Chronic kidney disease in aged cats: clinical features, morphology, and proposed pathogeneses

CA Brown, J Elliott, CW Schmiedt… - Veterinary …, 2016 - journals.sagepub.com

… complex glomerulonephritis have been described in FIV-positive cats. In addition to these
… noted in FIV-positive cats. However, in comparing FIV-infected and FIV-noninfected cats in …

Save Cite Cited by 196 Related articles All 11 versions

[PDF] researchgate.net

Risk factors associated with the development of chronic kidney disease in cats evaluated at primary care veterinary hospitals

JP Greene, SL Lefebvre, M Wang… - Journal of the …, 2014 - Am Vet Med Assoc

… Another study 8 of the potential association between FIV infection and CKD in cats revealed
no association between the 2 diseases but did find a significant association between FIV …

Save Cite Cited by 123 Related articles All 9 versions

[HTML] sagepub.comFull View

Risk and protective factors for cats with naturally occurring chronic kidney disease

K Piyarungsri… - Journal of feline medicine …, 2017 - journals.sagepub.com

… an association between infectious diseases and CKD in catsFIV infection has been found
… of renal tissues of cats experimentally and naturally infected with FIV showed renal changes, …

Save Cite Cited by 22 Related articles All 8 versions

So, from my way of thinking, the impact of long standing FIV is  a long erm decline in the CD4 count, not much in the way of immunosuppression related diseases (PCP, toxoplasmosis, cryptococcosis, MAC, etc) but the cats have a foreshortened life and die of CKD or lymphoma when they are about 10 years old. The problem – is that CKD and lymphoma are common diseases of old cats and common causes of death, so in many studies and for many deaths, unless FIV testing is routine, clinicians might miss this IMPORTANT association – because they just diagnose the lymphoma or CKD and MISS the underlying FIV infection. And this fits in what i have seen clinically. It would so good to see a prospective study looking at a much larger cohort kept in people houses (not a research facility) – I bet the strong association is between FIV and mortality, i.e. FIV positive cats don’t live as long as FIV negative cats, with earlier onset of the common diseases of the older cat (CKD and lymphoma).

Material has been added to the Discussion section regarding the possible etiologic link between FIV status and CRF.
